# Recurrent Vulvovaginal Candidosis and Its Underlying Mechanisms: A Systematic Review

**DOI:** 10.3390/jof11050357

**Published:** 2025-05-05

**Authors:** Maria Lobo, Catarina Cerqueira, Acácio Gonçalves Rodrigues, Carmen Lisboa

**Affiliations:** 1RISE-Health, Department of Pathology, Microbiology, Faculty of Medicine, University of Porto, 4200-319 Porto, Portugal; up201906394@med.up.pt (M.L.); agr@med.up.pt (A.G.R.); 2Department of Dermatology and Venereology, ULS Braga, 4710-243 Braga, Portugal; catarina.dias.cerqueira@gmail.com; 3Department of Dermatology and Venereology, ULS São João, 4200-319 Porto, Portugal

**Keywords:** recurrent vulvovaginal candidosis, vulvovaginal candidosis, recurrence mechanisms, chronic yeast infections, Candida species, systematic review

## Abstract

Recurrent vulvovaginal candidosis (RVVC), defined as three or more episodes of vulvovaginal candidosis (VVC) within a 12-month period, is a common and debilitating condition that affects a significant percentage of reproductive-aged women, negatively impacting their quality of life. This review aimed to synthesize the most recent scientific data on the pathophysiological mechanisms triggering primary or idiopathic RVVC. Three databases (PubMed, Scopus, and Web of Science) were searched for studies published between 2014 and 2024. Twelve studies were included, covering prospective cohort, cross-sectional, and case–control studies conducted in different countries. The results indicate that recurrence may be related to both intrinsic characteristics of the pathogen, such as increased virulence attributes of *Candida* spp., and host immune system dysregulation, including alterations in Th1/Th17 and Th2/Treg cytokine levels, decreased levels of mannose-binding lectin (MBL), impaired neutrophil and lymphocyte function, and overexpression of CD163+ macrophages and NLRP3 inflammasome. Additionally, genetic factors, such as polymorphisms in the *MBL2*, *IL-12*, *NLRP3*, and *TLR2* genes, are associated with increased susceptibility to RVVC. In conclusion, RVVC appears to involve a complex interaction between pathogen virulence and an altered host immune response, which reinforces the need for further investigations to develop personalized therapeutic strategies.

## 1. Introduction

Vulvovaginal candidosis (VVC) is a fungal infection caused by *Candida* species characterized by inflammation of the vulval and vaginal epithelium. It is a very common and distressing condition, affecting up to 75% of women at least once in their lifetime, most frequently during their reproductive years [1]. Defined as three or more episodes of VVC within a 12-month period, recurrent vulvovaginal candidosis (RVVC) represents approximately 5–8% of all cases of VVC in reproductive-aged women [2]. *Candida albicans* is the primary causative agent of acute sporadic VVC and, by extension, most cases of RVVC, although non-*C. albicans* species such as *Nakaseomyces glabratus* (previously known as *Candida glabrata*) are increasingly being implicated in recurrent episodes [3].

Women with RVVC tend to experience higher levels of psychological distress compared to those without a history of the condition. This includes increased susceptibility to depression, anxiety, and diminished self-confidence [4,5,6]. Furthermore, RVVC significantly impacts their emotional and sexual relationships and may contribute to reduced productivity and financial distress [4,5,6].

Patients often report dysuria, vulvar pruritus, dyspareunia, and soreness; signs include vulvar edema and erythema, fissures, excoriations, and vaginal discharge, together with introital and vaginal erythema [2,3]. While these symptoms and signs are characteristic, they are not exclusive to candidosis [3]. Therefore, diagnosis should not rely solely on clinical evaluation. It must also include either a wet preparation of vaginal discharge to identify budding yeasts, hyphae, or pseudohyphae (using saline and 10% KOH microscopy) or culture, confirming the presence of a yeast species. In cases of RVVC, culture of a vaginal exsudate or polymerase chain reaction (PCR) is essential to identify *Candida* species [2].

The adherence of *Candida* spp. to vaginal epithelial cells is followed by the colonization of the vagina, which is known to be enhanced by an estrogen-influenced environment. In healthy women not particularly susceptible to RVVC, asymptomatic colonization may persist for months or years, since yeasts live in symbiosis with the vaginal microbiota. Acute symptomatic VVC episodes usually follow a breakdown in this relationship and entail either a triggered overgrowth of *Candida* species or impairment of the host protective defense mechanisms, particularly the innate immune response, which serves as the first line of defense. The causes of this disruption in homeostasis vary and may be related to sexual intercourse, the use of contraceptives, antibiotic use, and uncontrolled diabetes [1]. Nevertheless, primary or idiopathic RVVC, where no secondary causes or triggers are identified, does occur [3].

Regarding treatment of RVVC, the approach typically involves 7–14 days of induction therapy with a topical or oral antifungal agent, followed by weekly maintenance (oral fluconazole) for 6 months. However, the treatment strategy depends on the specific yeast/*Candida* species present and the azole susceptibility of the isolates. This is especially critical, as *C. albicans* azole resistance is becoming more common among vaginal isolates, and some non-*C. albicans* species are intrinsically resistant to azoles [2,7]. Although suppressive maintenance therapies are effective at controlling RVVC, they are rarely curative in the long term. In fact, several studies point to the recurrence of episodes in a significant percentage of patients after the cessation of maintenance therapy in the absence of an apparent triggering event [8,9,10,11,12,13,14].

A deeper understanding of the pathophysiology underlying primary or idiopathic RVVC, which remains significantly limited, will certainly enable the development of more effective treatment strategies. This, combined with the significant toll on the overall well-being of those affected, warrants more in-depth study of the subject. This review aims to synthesize the latest available scientific data on the pathophysiological mechanisms triggering primary or idiopathic RVVC.

## 2. Materials and Methods

### 2.1. Search Strategy

This systematic review was performed according to the guidelines of the Preferred Reporting Items for Systematic Reviews and Meta-Analyses (PRISMA) 2020 checklist [15]. The PubMed, Scopus, and Web of Science databases were searched for studies published between 2014 and 2024. The literature search was performed on 21 October 2024. The combination of title words, abstract words, and keywords included (Recurrent OR Persistent OR Chronic OR Recurrence OR Complicated) AND (“vulvovaginal candidiasis” OR “vaginal candidiasis” OR “vulvovaginal candidosis” OR “vaginal candidosis” OR “candida vaginitis” OR “candida vulvovaginitis”). Covidence Systematic Review Manager was used to collect all the research results and to select the studies. The titles and abstracts of the articles were reviewed by two independent and blinded reviewers to determine potential eligibility. Full publications were reviewed for subsequent inclusion. Any conflicts were resolved through discussion with one other reviewer. The details of the search strategy are provided in Appendix A.

### 2.2. Selection Criteria

Studies were eligible if they evaluated a potential mechanism of RVVC by comparing RVVC-affected women with women with non-recurrent VVC and/or healthy women, regardless of their age. Additionally, a diagnosis of RVVC had to be clearly defined as three or more episodes of VCC within a 12-month period. The exclusion criteria were women diagnosed with other causes of vaginitis (e.g., bacterial vaginosis or trichomoniasis), diabetes mellitus, or immunosuppressant conditions such as HIV infection or iatrogenic immunosuppression, as well as those with a clinical diagnosis only. Studies regarding quality of life; virulence and/or host factors not related to recurrence; and efficacy/efficiency, safety, and/or cost-effectiveness of treatments were also excluded.

### 2.3. Data Extraction and Quality Assessment

A standardized protocol established at the start of the literature search was used to summarize data from eligible publications. Extracted information included the first author’s name and year of publication, study design, country or region where the study was conducted, the aim of the study, study population, comparator, *Candida* spp. identification method, Antifungal Susceptibility Testing (AST) method, genetic study, immunological study, collected samples, results/key findings, limitations, and quality assessment. Data were reviewed by three researchers (M.L., C.L., and C.C.). Furthermore, blinded quality assessment was performed independently and blinded by the same three researchers (M.L., C.L., and C.C.) using the National Institutes of Health quality assessment tools [16]. Divergences were discussed among the researchers.

## 3. Results

Our search identified a total of 1010 articles. After removing 533 duplicates, 416 articles were also excluded based on title and abstract analysis. The remaining 61 full-text articles were assessed for eligibility, resulting in the exclusion of 49 studies for the following reasons: 3 reported outcomes not relevant to our research question, 1 included an inappropriate control population (comparator), 16 did not follow an eligible study design (review), 3 were available only in a non-English language, 5 involved an incorrect patient population (misdefined RVVC), and 21 did not fully meet the exclusion criteria (Figure 1).

### 3.1. Study Design, Site, and Population

This systematic review included a total of twelve studies: two prospective cohort studies [17,18], two cross-sectional studies [19,20], and eight case–control studies [21,22,23,24,25,26,27,28]. Four studies were conducted in China [17,18,23,26], two in Iran [21,24], and two in Colombia [19,20]. Additionally, one study [27] included participants from in the USA, the Netherlands, and France, while another [25] enrolled participants from these three countries as well as Italy. The remaining two studies were conducted in Egypt [22] and Turkey [28].

Altogether, the aforementioned studies gathered data from 2277 women: 843 diagnosed with RVVC, 206 diagnosed with VVC, and 1228 healthy women (with no diagnosis of either VVC or RVVC). The cohort of participants in the two Colombian studies [19,20] is the same and included women aged between 18 and 50 years and 40 RVVC patients. Two of the Chinese studies included female cohorts aged 15–54 and 18–53 years old, with populations of RVVC-diagnosed patients of 20 and 48 women, respectively [17,23]. The study conducted in Egypt [22] included 59 women diagnosed with RVVC aged between 30 and 40 years. Three of the studies [21,24,28] did not report the age range of the participants but, instead, provided the mean age of the RVVC patient group and the mean age of the corresponding control group: in one of two the studies conducted in Iran, the mean age of the patient and control groups was 36.18 ± 10.26 and 36.43 ± 9.98 years, respectively, with a total of 20 RVVC patients [21], while in the other, the mean age of the patient and control groups was 23.4 ± 6 and 25.8 ± 7 years, respectively, with a total of 95 RVVC patients [24]. Similarly, in the study conducted in Turkey, the mean age of the patient and control groups was 29.06 ± 7.3 and 31.41 ± 6.2 years, respectively, with a total of 50 RVVC patients [28]. In the studies conducted in France, the Netherlands, and the USA, the only information provided regarding participants’ age was that all were at least 18 years old [25,27]. Regarding the number of included RVVC-diagnosed women, one study enrolled 270 [25], while the other enrolled 119 [27]. Contrary to the remaining studies, two studies conducted in China did not provide information on the participants’ age [18,26]. One of them included 58 women diagnosed with RVVC [26], while the other included 44 women with the same condition [18].

Five studies [17,21,22,23,28] found no statistically significant difference in age between cases and controls; the remaining seven did not mention this potential difference. None of the twelve studies unequivocally reported a statistically significant age difference between cases and controls.

Two studies [22,24] excluded women using hormonal contraceptives. Three studies [19,20,28] included participants using hormonal contraceptives, albeit without a statistically significant difference between cases and controls, whereas others [17,18,21,23,25,26,27] did not mention excluding such participants.

The multicenter case–control studies [25,27] using healthy women as controls enrolled RVVC patients during episodes of acute vaginitis or, if asymptomatic, while receiving maintenance fluconazole therapy. Three studies [17,19,20] excluded women under antifungal treatment at least 7 days prior to vaginal sample collection. The remaining studies did not mention whether participants were currently taking antifungals or had a history of antifungal use.

The characteristics of the women included in the comparator group differed between studies. Eight studies [19,20,21,22,24,25,27,28] considered healthy women as the control group, while a single study included women diagnosed with VVC as the unique comparator [17]. Three studies used both [18,23,26]. One study [18], in addition to VVC patients and healthy women, also used genotypes of multiple isolates from the same patient as a comparator. Table 1 shows the main characteristics of the included studies.

### 3.2. Method for Candida spp. Identification

As previously mentioned, all studies included only cases of RVVC in which the diagnosis had been microbiologically confirmed. However, the method by which *Candida* spp. was identified (either exclusively for diagnostic purposes or as an integral part of the research methodology) varied among studies.

In one study [28], diagnosis was confirmed using vaginal discharge cultures collected from the posterior fornix, although the method by which species distinction was performed was not specified. Three studies [19,20,22] employed a biochemical method for the identification of *Candida* spp.; two of them used the Vitek^®^ 2 Compact system (BioMérieux, Inc., Durham, NC, USA) after vaginal fluid samples were cultured on Sabouraud dextrose agar (SDA) medium (Scharlau, Barcelona, Spain) and *Candida* Brilliance agar medium (Oxoid, Basingstoke, UK) [19,20], and the other used the Hi-*Candida* API identification kit (BioMérieux, France) after vaginal fluid samples were culture on SDA medium (Oxoid, UK) and chromogenic agar medium (CHROMagar™ Candida, Paris, France) [22]. In two studies [18,26], *Candida* spp. was identified using the VITEK MS automatic microbial mass spectrometry system. Three studies employed molecular methods, specifically sequencing of the Internal Transcribed Spacer (ITS) region of DNA [17], Fluorescent In Situ Hybridization (FISH) [23], and PCR-Restriction Fragment Length Polymorphism (PCR-RFLP) [24]. In one study, identification was performed simultaneously by colony color assessment in chromogenic agar medium (CHROMagar™ Candida, Paris, France) and sequencing of the ITS region of DNA [21]. Two of the studies [25,27], despite confirming that RVVC diagnosis was not based solely on clinical criteria, did not specify the method used to identify *Candida* species.

### 3.3. Results/Key Findings

#### 3.3.1. Recurrence Related to Pathogen Characteristics

One of the studies [17] aimed to investigate the potential impact of differences in *Candida* species distribution between RVVC and VVC patients on infection recurrence. The study found that *C. albicans* accounted for 90% and 96.1% of all strains collected from RVVC and VVC patients, respectively. The proportion (10%) of non-*C. albicans Candida* (NAC) species was higher in the RVVC group than in the VVC group (3.9%), but there was no statistical difference (*p* > 0.05).

Another study [18] sought to determine whether the recurrence of VVC results from relapse caused by the same pathogen or reinfection caused by a new pathogen. In this study, genomic DNA of *C. albicans* was extracted, and seven genes (AAT1a, ACC1, ADP1, MPIb, SYA1, VPS13, and ZWF1b) were amplified and sequenced. The MLST (Multilocus Sequence Typing) of these strains assigned each a specific DST (Diploid Sequence Type). The results showed that 59.1% (26/44) of patients experienced a relapse, with DST 79 (65.4%) as the dominant genotype. The etiology of the remaining 40.9% (18/44) of patients was reinfection, with the main genotypes including DST 79 (33.3%), DST 124 (8.6%), and DST 1895 (8.6%).

A comparison of antifungal susceptibility between RVVC and VVC patients was also performed in one of the studies [17]. The tested antifungals included voriconazole, posaconazole, amphotericin B, terbinafine, ketoconazole, fluconazole, itraconazole, and micafungin. The study revealed that the antifungal susceptibility profiles of the RVVC group were consistent with those of the VVC group.

Another study [26] aimed to assess the role of *C. albicans* virulence in RVVC recurrence compared to antifungal susceptibility. The germ-tube formation rate and biofilm formation ability of strains in the RVVC group were significantly higher than those of strains in the VVC and healthy groups (*p* < 0.001). In this regard, higher virulence was observed in RVVC isolates compared to VVC patients and healthy women. Furthermore, the antifungal susceptibility of strains in the VVC group was significantly lower than that of strains in the RVVC and healthy groups.

Conseguegra-Asprilla J.M. et al. [20] performed antifungal susceptibility tests in *Candida* spp. isolated from women with RVVC. Four fluconazole-resistant *C. albicans* strains were found, and two isolates were resistant to voriconazole. Regarding *C. lusitaniae*, one isolate was resistant to 5-flucytosine.

Table 2 shows antifungal susceptibility tests performed regarding *C. albicans* in RVVC patients.

#### 3.3.2. Recurrence Related to Immune System Dysregulation

Six studies [17,19,20,21,22,23] evaluated the immune response of RVVC patients compared to the respective controls.

In one of them [17], the serum levels of Interferon (IFN)-γ, Tumor Necrosis Factor (TNF)-α, and Interleukin (IL)-17F in the RVVC group were lower, whereas those of IL-4, IL-6, and IL-10 were higher when compared to the VVC group, with statistical significance (*p* < 0.05). Therefore, RVVC patients exhibited lower amounts of T-helper (Th)1 and Th17 cytokines but a higher amount of Th2 cytokines when compared to VVC patients. In another study [19], the expression of genes associated with different immunological profiles was determined in situ, i.e., in the vaginal mucosa (not in the serum). In patients with RVVC, decreased expression of T-bet, RORγ-T, IL-1β, and IL-17, as well as increased expression of FOXP3, IL-4, IL-8, IL-10, and IL-18, was observed when compared to healthy women (*p* < 0.05). Therefore, RVVC patients exhibited decreased Th1/Th17 and increased Th2/Treg cytokines in the vaginal mucosa.

The influence of serum levels of Mannose-Binding Lectin (MBL) on VVC recurrence was evaluated in three studies [19,21,22]. One of them found decreased levels of MBL in RVVC patients (*p* < 0.01) compared to healthy controls [19]. In another study, no statistically significant difference in MBL serum levels was observed between RVVC cases and healthy women (*p* = 0.145) [22]. However, another study revealed that the MBL concentration was significantly higher in participants suffering from RVVC compared to healthy women (0.330 ng/mL vs. 0.253 ng/mL) (*p* = 0.001) [21].

One study showed a significant decrease in the fungicidal capacity of polymorphonuclear neutrophils (PMNs) from RVVC patients when compared to healthy women (*p* < 0.05) [20]. Peripheral blood mononuclear cells (PBMCs) from RVVC patients stimulated with *C. albicans* exhibited a significant decrease in the proliferation index when compared to PBMCs obtained from healthy women stimulated with the fungus (*p* < 0.01) [20].

CD163+ macrophages and pyrin domain-containing protein 3 (NLRP3) inflammasome overexpression were identified in vaginal mucosa biopsies from RVVC patients when compared to VVC patients and healthy women (*p* < 0.01) [23].

#### 3.3.3. Recurrence Related to Genetic Factors

The association between the *MBL2* gene exon 1 codon 54 polymorphism and susceptibility to RVVC was assessed by Hammad N.M. et al. [22]. Allele A (wild-type allele) was present in 83.9% of RVVC cases and in 94.0% of healthy controls, whereas allele B (mutant allele) was present in 16.1% of RVVC cases and in 6% of controls. The distribution of *MBL* genotypes (AA, AB, and BB) and alleles differed significantly between RVVC cases and controls (*p* = 0.038 and 0.013, respectively). The risk of RVVC was 3.04 times higher among carriers of allele variant “B” compared to those who were not carriers. In the RVVC group, serum MBL levels in mutant genotypes (AB and BB) were significantly lower than in the wild type (AA) (*p* = 0.019 and 0.033, respectively). However, in the control group, no statistically significant difference was found between the mutant and wild-type MBL serum levels (*p* = 0.23).

One study [24] evaluated the effect of genetic variants of *IL-12* genes, as well as IL-12 mRNA expression levels, on susceptibility to RVVC. The expression level of the *IL-12* gene in patients was twofold higher than that of the healthy women (*p* = 0.0013). Regarding a single-nucleotide polymorphism (SNP) in *IL-12*, three genotypes were identified at the rs568408 position (AA, AG, and GG). The ratio of *IL-12A* genotypes among RVVC patients was 29.47% GG, 63.15% AG, and 7.38% AA, while for the control group, it was 6.94%, 73.26%, and 19.80%, respectively. There was a significant difference in genotype distribution between the patient and control groups (*p* < 0.001). Additionally, the analysis of *IL-12A* gene alleles at the rs568408 position revealed that 38.94% of patients carried allele A, while 61.06% carried allele G, versus 43.57% of healthy individuals with allele A and 56.43% with allele G. Significant differences in the frequencies of A and G alleles were observed between the patient and control groups (*p* < 0.001). Therefore, the results of this case–control study suggest that genetic variation in *IL-12* may significantly enhance susceptibility to RVVC by upregulating IL-12 expression.

Jaeger M. et al. [25] assessed the role of a variable-number tandem repeat (VNTR) polymorphism in the *NLRP3* gene in patients suffering from RVVC, as well as the functional consequences of this sequence variant for specific host defense mechanisms against *C. albicans* in the vaginal fluid. The 42 bp VNTR within intron 4 of the *NLRP3* gene resulted in eight genotypes derived from five different alleles. One of these genotypes, 12/9, was significantly more prevalent among RVVC patients than in controls (*p* = 0.02). IL-1β levels were higher in the vaginal fluid of RVVC patients compared to healthy controls. In addition, the 12/9 genotype was associated with higher IL-1β concentrations in the vaginal fluid compared to the 12/12 genotype (wild type), suggesting that IL-1β-mediated hyperinflammation driven by the *NLRP3* gene plays a role in RVVC pathogenesis.

The impact of SNPs in the genes coding for *DECTIN-1*, *CARD9*, *TLR1*, *TLR2*, and *TLR4* on susceptibility to RVVC was studied in [27]. While *CLEC7A*, *CARD9*, *TLR1*, and *TLR4* polymorphisms had no apparent impact, the *TLR2* Pro631His polymorphism (due to the substitution of proline by histidine) was associated with a 2.705-fold increase (*p* = 0.046) in susceptibility to RVVC. Furthermore, this *TLR2* SNP had deleterious effects on protein function, as assessed by in silico analysis. In vitro functional assays suggested that it reduces IL-17 and IFN-γ production upon stimulation of peripheral blood mononuclear cells with *C. albicans*.

Finally, another study [28] investigated whether the Human *Dectin-1 Y238X* gene polymorphism plays a role in RVVC pathogenesis. When comparing three different *Dectin-1* genotypes, no significant difference was found between the two groups (RVVC and controls) (*p* = 0.452, *p* = 0.615, and *p* = 0.275). However, a significantly higher proportion of RVVC patients had a family history of RVVC (*p* = 0.001). In fact, in another study, 17.5% of RVVC patients reported having relatives up to the third degree of consanguinity with a previous history of VVC (OR = 5.3; *p* = 0.024) [20].

### 3.4. Quality Assessment

Among the twelve studies included in this review, the quality of ten was considered to be fair, and two were considered to be of good quality according to the NIH tools [16].

All the studies had limitations concerning a small sample size, a lack of sample-size justification, random sampling of the participants, and blinding of exposure assessors. Some studies had limitations concerning the definition of the study population [17,19,20,24,28] and the definition and origin of control cases [25,27].

The prospective cohort study by Tian et al. [18] and the case–control study conducted by Hammad et al. [22] were considered to be of good quality because they scored positively concerning the definition of the population, inclusion and exclusion criteria, and the definition of exposure/risk and outcome. Additionally, the study conducted by Tian et al. [18] assessed three different groups—RVVC patients, VVC patients, and healthy women—and was the only study that analyzed multiple isolates from the same patient in different infectious periods. The study conducted by Hammad et al. [22] was the only study that matched cases and controls for age, marital status, and socioeconomic status.

## 4. Discussion

As previously mentioned, the purpose of this review was to assemble the most recent findings regarding the mechanisms through which primary or idiopathic RVVC recurs. Upon analyzing the obtained results, we found that they can be classified into two groups: those related to the intrinsic properties of the pathogenic agent and those associated with host characteristics that predispose to increased susceptibility.

Clinically, any reappearance of the disease may be the result of reinfection or persistence [29]. The former is related to conditions that render the host more vulnerable, such as the use of antibiotics, immunosuppressants, or dysfunction of the host immune system. The latter often results from a failure of antifungal treatment due to resistance to antifungals or the ability of *Candida* spp. to produce biofilms with biochemical properties that may be difficult to eradicate. In addition to biofilm formation, *Candida* spp. can express other virulence attributes, such as the formation of germ tubes and hyphae [30]. One study specifically compared the frequency of relapse (persistence) versus reinfection. The results showed that 59.1% of RVVC patients experienced a relapse, whereas the etiology of the remaining 40.9% was reinfection, suggesting that reinfection and persistence occur at similar frequencies [18]. Regarding the mechanisms related to persistence, two of the studies included in this review were consistent in their findings in terms of antifungal resistance, concluding that there was no significantly higher antifungal resistance among RVVC patients [17,26]. Interestingly, it was demonstrated that higher expression of virulence attributes may play a more significant role in persistence than acquired resistance to antifungals [26]. However, the lack of knowledge regarding participants’ history of antifungal use may have contributed to potential bias in these results. As previously stated, the prevalence of non-*C. albicans Candida* species continues to increase globally, and these species are more commonly found among RVVC patients [3], a fact often attributed to their propensity for lower antifungal susceptibility [31]. Nevertheless, in the single study comparing their frequency between RVVC and VVC patients, although the proportion of non-*C. albicans* species was, indeed, higher in the RVVC group (10% vs. 3.9%), there was no statistically significant difference between the two groups [17]. It is important to note that the generalizability of these results may have been compromised by the limited geographic scope of the studies, as the relative prevalence of *C. albicans* and non-*C. albicans* species can vary across regions due to environmental differences, hygiene practices, and cultural factors.

Regarding host defense mechanisms, the innate immune system provides the first barrier against vulvovaginal candidosis. Pattern recognition receptors (PRRs) on innate immune cells detect molecular moieties on the yeast surface, triggering intracellular signals within epithelial cells that stimulate the production of effector molecules. Toll-like receptors (TLRs) are an example of PRRs and recognize numerous components of fungal cell walls. Single-nucleotide polymorphisms in host genes have been associated with increased susceptibility to candidosis [2]. One of the studies included in this review demonstrated precisely this, as a specific polymorphism in the TLR2 gene was simultaneously associated with increased susceptibility to RVVC and decreased receptor function [27]. This loss of function was linked to reduced levels of IFNγ and IL-17, indicating a failure in the activation of the Th1/Th17 response, which is essential in controlling fungal infections [19]. Dectin-1 is another PRR belonging to the C-type lectin family whose mutation has been linked to susceptibility to RVVC [3]. However, in one of the studies included in this review, no association was found between a *dectin-1* polymorphism and increased susceptibility to RVVC. Screening in larger populations may affect these results [28].

Another component of innate immunity is MBL, a soluble receptor that binds to *Candida* species and activates complement, enhancing opsonization [3]. One study determined that the MBL concentration was significantly higher in RVVC patients compared to the control group [21]. This finding is consistent with reports by Henic et al., who also found that MBL serum levels were higher in RVVC cases compared to controls, suggesting that MBL may provide a defense against RVVC [32]. Conversely, another study included in this review [19] detected decreased MBL levels in RVVC patients compared to healthy controls, aligning with research conducted by Babula et al. [33]. The association between an *MBL2* exon 1 gene polymorphism and susceptibility to RVVC was also evaluated, showing that the risk of RVVC was approximately three times higher among individuals carrying a specific allelic variant compared to those who did not. Among the RVVC patient group, this polymorphism resulted in lower MBL serum levels [22]. Similarly, Liu et al. [34] and Namarat Kalia et al. [35] reported that alterations in the *MBL* gene lead to decreased MBL serum levels and recurrent infections. Contrary to what was previously stated, these findings suggest that recurrence may be due to decreased MBL levels, which, in turn, reduce vaginal host defenses against *Candida* species. The discrepancies among different studies could be related to differences in ethnicity, sample size, and study methodologies. The efficacy of MBL protein as a potential therapeutic agent against RVVC, particularly among MBL-deficient women, should be further investigated.

CD163+ macrophage overexpression was identified in biopsies of vaginal mucosa of RVVC patients when compared to VVC cases and healthy women [23]. CD163 is a surface receptor that is highly upregulated and selectively expressed in M2 macrophages. On the cell membrane, its function is to participate in the Th2-type immune response, inhibiting the Th1-type immune response and promoting local immune tolerance [23]. Therefore, the presence of a large number of M2 macrophages in the vaginal submucosa may result in vaginal immune tolerance, failing to efficiently eliminate the yeasts, resulting in a state of continuous damage and incomplete repair of vaginal tissue [23].

Regarding adaptive immune response, NLRP3 is an intracellular receptor that is part of a multicomplex cytoplasmic protein platform called the inflammasome. The inflammasome enables the activation of caspase-1, which, in turn, cleaves the inactive precursors of IL-1β and IL-18 into their biologically active cytokine forms [25]. In one of the studies included in this review [23], NLRP3 overexpression was identified in RVVC cases (vaginal mucosa biopsies) compared to VVC patients and healthy women. Additionally, another study [25] concluded that a polymorphism in the *NLRP3* gene was associated with increased susceptibility to RVVC and higher levels of IL-1β in vaginal fluid, which were also elevated in the RVVC group. This suggests that IL-1β-mediated hyperinflammation, driven by the *NLRP3* gene, plays a putative role in RVVC pathogenesis [25]. Supporting these findings, current models of VVC pathogenesis suggest that epithelial activation by *Candida* spp. leads to the production of inflammatory mediators, namely IL-8 and IL-1β, through the NLRP3 inflammasome and the recruitment of PMNs. However, these cells may be ineffective in reducing fungal burden [36]. In fact, one of the analyzed studies showed a significant decrease in the fungicidal capacity of PMNs in RVVC patients compared to healthy women [20]. PMNs, despite being more numerous in RVVC patients [20], exhibit reduced fungicidal activity due to “neutrophil anergy”, allowing *Candida* spp. to persist in the vaginal tract, as demonstrated by Yano et al. [36]. It is worth noting that IL-17/Th 17 plays a crucial role in neutrophil recruitment and activation. *Candida* spp. killing by mononuclear leukocytes from Th17 inhibitor recipients with a history of candidiasis was impaired compared to healthy controls in a real-world observational study [37].

Furthermore, PBMCs from RVVC patients stimulated with *C. albicans* revealed a significant decrease in the proliferation index compared to PBMCs from healthy women stimulated with the fungus [20]. This result aligns with the findings of Corrigan et al. [38] and Talaei et al. [39], who observed that RVVC patients exhibited reduced T-lymphocyte proliferation indices in response to *Candida* spp. Likewise, a study conducted by Nawrot et al. [40] reported that PBMCs isolated from RVVC patients displayed low proliferation rates when exposed to a *Candida* antigen. These findings suggest that the adaptive immune response may be impaired in RVVC patients.

Still concerning the adaptive immune response, CD4+ T cells and their cytokines play a central role in antifungal immunity at the mucosal surface. Th1 and Th17 cells are the main effector cells responsible for protective immunity against fungi, whereas Th2 responses are thought to be associated with various deleterious effects, including by exerting an antagonistic effect by suppressing cellular (Th1-type) immunity [19]. Our data support the hypothesis that T-cell immunity could provide protection against recurrent fungal vaginitis, as demonstrated by the differences in cytokine responses between RVVC and controls. The immune responses of VVC patients favored the activation of cytokines produced by Th1 and Th17 cells, which were significantly less abundant in RVVC patients. In contrast, cytokines associated with Th2 and Treg cells were more abundant in RVVC patients, suggesting a likely Th2 response in RVVC [17,19]. Genetic differences between populations may influence infection susceptibility and immune responses, potentially hindering the extension of these results to other ethnic or genetic groups.

IL-17 inhibitors are biologics currently used to reduce inflammation and tissue injury associated with chronic inflammatory diseases (like psoriasis and ankylosing spondylitis) by targeting the T-helper-17 pathway. These drugs are associated with an increased risk of candidosis, including VVC. In fact, observational data and clinical trials show an increased incidence of candidosis in individuals treated with IL-17 inhibitors [37,41].

Furthermore, the immune response against *Candida* species is compromised when IL-17 is also inhibited through genetic knockout [42].

In this era of antifungal resistance, the search for an individualized immunomodulatory treatment that enhances and improves the host’s immunity in the fight against *Candida* spp. infections is an urgent need. Anti-*Candida* spp. vaccines are under development, and promising results are awaited. With advances in the understanding of the immunopathogenesis of RVVC and host-related factors, this challenge is becoming more tangible and promising.

This systematic review has some limitations, many of which are related to the included studies. The overall findings are based on a small sample size, and many of the studies lack generalizability. Additionally, the studies show variability regarding the control group and inconsistency related to antifungal treatment before sample collection. Moreover, three of them did not clearly specify the microbiological method used to diagnose vulvovaginal candidosis [25,27,28]. The age distribution across studies is relatively consistent, although two studies did not provide information on the participants’ age [18,26] and two others only stated that the included women were over 18 years old [25,27].

Five studies [17,21,22,23,28] found no statistically significant difference in age between cases and controls; the remaining seven did not mention this potential difference. In three studies [19,20,28], participants using hormonal contraceptives were included, albeit without a statistically significant difference between cases and controls, whereas others [17,18,21,23,25,26,27] did not mention excluding such participants. Since estrogen levels vary with age and confer greater susceptibility to infection, the absence of a significant difference between cases and controls regarding age and the use of hormonal contraceptives is necessary for valid conclusions to be drawn without the influence of this potential confounding variable.

Another limitation of this review was incomplete access to identified research, as three of the identified articles were written in Chinese.

## 5. Conclusions

The clinical reappearance of vulvovaginal candidosis in an RVVC patient may result from reinfection or persistence. Regarding persistence, the data indicate a predominant role of virulence attributes rather than antifungal resistance. Alterations in the host’s innate and adaptive immune responses are, in turn, responsible for an increased predisposition to reinfection through an exacerbated and ineffective chronic inflammatory response. These immune alterations may be caused by various mutations in genes encoding inflammatory mediators. Further deciphering of vaginal host defense mechanisms against *Candida* spp. is essential for the design of novel and personalized chemical and/or immunotherapeutic strategies that could specifically target the underlying immune defects in RVVC, aiming to improve and/or replace standard antifungal treatments.

## Figures and Tables

**Figure 1 jof-11-00357-f001:**
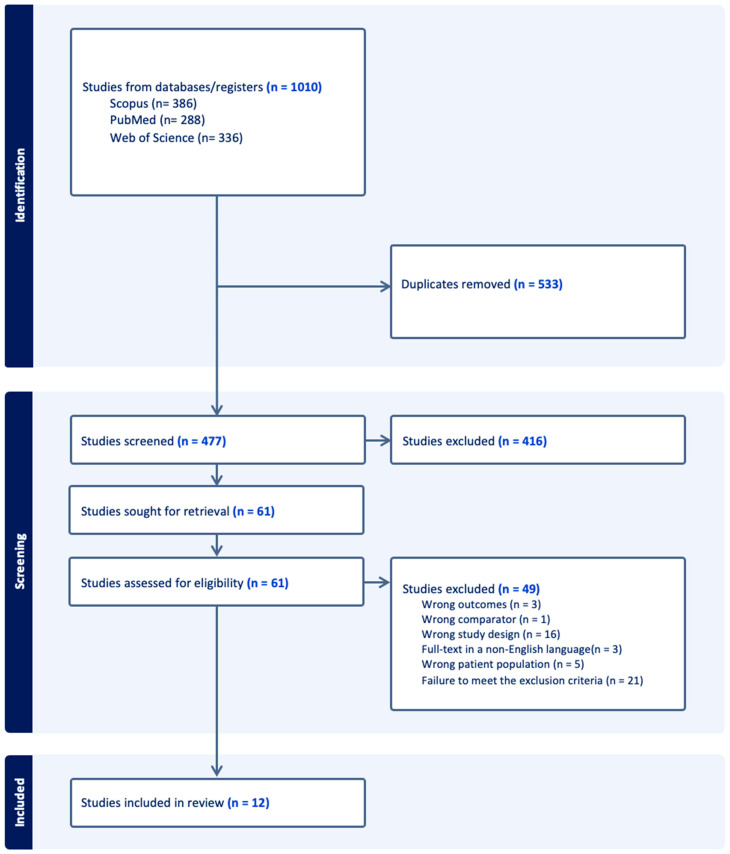
PRISMA flow chart.

**Table 1 jof-11-00357-t001:** Main characteristics of the included studies.

First Author, Publication Year	Study Design	Origin (Country)	Objective	Population (*n*)	Comparator	Method to ID *Candida* spp.	TSA Method	Molecular Method	Immunological/ Inflammation Method	Collected Sample	Results/ Key Findings	Limitations	Quality Assessment
Consuegra-Asprilla J.M. et al., 2024 [20]	Cross-sectional study	Colombia	Evaluate the fungicidal activity of PMNs and the proliferation of PBMCs in Colombian patients diagnosed with RVVC.	66 women, aged 18–50 years: 40 women with RVVC and 26 healthy women.	Healthy women	Vitek^®^2 Compact system.	Vitek^®^2 Compact system.	Not applicable.	PBMC proliferation assessed via flow cytometry. Neutrophil fungicidal activity calculated as the percentage reduction in CFUs.	Blood samples.	RVVC patients exhibited a significant decrease in the fungicidal capacity of the PMNs. PBMCs from RVVC patients stimulated with *C. albicans* showed a significant decrease in the proliferation index.	Small sample size; limited geographic scope. Study limited to *C. albicans*	Fair
Consuegra-Asprilla J.M. et al., 2024 [19]	Cross-sectional study	Colombia	Determine the expression, in the vaginal mucosa, of genes associated with different immunological profiles. Determine serum levels of vitamin D, dectin-1 and MBL.	66 women, aged 18–50 years: 40 women with RVVC and 26 healthy women.	Healthy women	Vitek^®^2 Compact system.	Vitek^®^2 Compact system.	Gene expression determined by qPCR.	Serum concentrations of dectin-1, MBL, and vitamin D assessed by ELISA.	1. Scraping samples of the vaginal mucosa (genetic analysis) 2. Blood samples (determination of vitamin D, dectin-1, and MBL levels).	RVVC patients exhibited decreased Th1/Th17 and increased Th2/Treg cytokines at the level of the vaginal mucosa. RVVC patients had decreased levels of MBL.	Small sample size; limited geographic scope.	Fair
Ge G. et al., 2022 [17]	Prospective cohort study	China	Investigate and compare the differences in *Candida* spp. distribution, antifungal susceptibility, and immune responses and cytokine levels between RVVC and VVC patients.	98 women, aged 15–54 years: 20 women with RVVC and 78 VVC patients.	VVC patients.	Molecular identification by sequencing the ITS region of DNA of *Candida* spp. strains.	Broth microdilution according to CLSI guidelines.	Not applicable.	Serum cytokine profiles measuredusing a bead-based multiplex assay (LEGENDplex™; BioLegend, San Diego, CA, USA).	1.Vaginal fluid samples (isolation, identification, and antifungal susceptibility testing of *Candida* species). 2. Blood samples (evaluation of cytokine levels).	*C. albicans* accounted for 90% and 96.1% of all strains isolated collected from RVVC and VVC patients, respectively, with no statistical difference. Antifungal susceptibility profiles of the RVVC group were consistent with those of the VVC group. RVVC patients had lower Th1 and Th17 cytokines but higher Th2 cytokines compared to VVC patients.	Small RVVC cohort; limited geographic scope.	Fair
Ghazanfari M. et al., 2019 [21]	Case–control study	Iran	Evaluate MBL levels and the relationship between the MBL serum level and the relative expression of MBL mRNA in RVVC patients.	80 women: 40 women with RVVC and 40 healthy women.	healthy women.	CHROMagar *Candida* medium. Molecular identification performed by sequencing the ITS region of DNA isolated from *Candida* spp. strains.	Not applicable.	mRNA expression of the MBL gene quantified using real-time qPCR.	Serum MBL levels measured via ELISA.	Blood samples.	The MBL concentration was significantly higher in participants suffering from RVVC compared to the control group. In the samples with significant upregulation of MBL mRNA expression, the results of the MBL serum level suggested that the MBL gene expression profile does not reflect phenotypic levels in the serum.	Small sample size; limited geographic scope.	Fair
Hammad N.M. et al., 2018 [22]	Case–control study	Egypt	Investigate the association between *MBL 2* gene exon 1 codon 54 polymorphism and susceptibility to RVVC in childbearing women.	118 women aged 30–40 years: 59 women with RVVC and 59 healthy women.	Healthy women.	Hi-Candida API identification kit for species identification.	Not applicable.	*MBL*2 exon 1 polymorphism analysed via PCR-RFLP.	Serum MBL levels measured by ELISA.	Blood samples.	No statistically significant difference in MBL serum level was observed between RVVC cases and controls. The distribution of MBL genotypes (AA, AB, and BB) and alleles significantly differed between RVVC cases and controls. The risk of RVVC was 3.04 times higher among those who carried allele variant “B”.	Limited to a single genetic variant; small sample size; limited geographic scope.	Good
He X. et al., 2023 [23]	Case–control study	China	Evaluate vaginal mucosal inflammation, including CD 163+ macrophages and NLRP3 expression.	144 women, aged 18–53 years: 48 women with RVVC, 48 VVC patients, and 48 healthy women.	VVC patients and healthy women.	Vaginal candidosis identified by FISH with ribosomally based probes.	Not applicable.	Not applicable.	CD163+ macrophages and NLRP3 expression determined by immunohistochemistry.	Vaginal biopsy tissues.	CD163+ macrophage levels were significantly higher in the RVVC group. The positivity rate of NLRP3 in RVVC patients was the highest among the three groups.	Small sample size; limited geographic scope.	Fair
Isakhani S. et al., 2022 [24]	Case–control study	Iran	Evaluate the effect of genetic variants of *IL-12* genes, as well as the mRNA expression level, on susceptibility to RVVC caused by *C. albicans.*	196 women: 95 women with RVVC and 101 healthy women.	Healthy women.	Vaginal candidosis confirmed by PCR-RFLP.	Not applicable.	Expression level and prevalence of polymorphisms in the *IL-12* gene assayed using real-time PCR and ARMS-PCR, respectively.	Not applicable.	Blood samples.	The expression level of *IL-12* gene in patients was two folds higher than that of the controls. A polymorphism in the *IL-12* gene was associated with susceptibility to RVVC.	Small sample size; limited geographic scope.	Fair
Jaeger M. et al., 2016 [25]	Case–control study	Italy, USA, Netherlands, and France	Assess the role of a polymorphism in the *NLRP3* gene in patients suffering from RVVC and investigate the functional consequences of this sequence variant on the specific host defense against *C. albicans* on the vaginal surface.	853 women of at least 18 years of age: 270 women with RVVC and 583 healthy women.	Healthy women.	Not specified.	Not applicable.	Genotyping of VNTR in the *NLRP3* gene performed using conventional PCR, followed by electrophoresis.	Cytokine levels in vaginal fluids determined by ELISA.	1.Blood samples (analysis of the *NLRP3* polymorphism). 2. Vaginal samples (determination of vaginal cytokine levels).	The expression of the 12/9 genotype was significantly higher in RVVC patients when compared with controls. IL-1β levels were higher in the vaginal fluid of RVVC patients compared to healthy controls. The 12/9 genotype led to even higher IL-1β concentrations compared to the 12/12 genotype (wild type).	Limited to specific VNTR variations; functional consequences need further study.	Fair
Li X. et al., 2022 [26]	Case–control study	China	Assess the role of *C. albicans* virulence in VVC recurrence compared to antifungal susceptibility.	127 women: 58 women with RVVC, 40 VVC patients, and 29 healthy women.	VVC patients and healthy women.	*C. albicans* identified by VITEK MS automatic microbial mass spectrometry system.	ATB FUNGUS 3 test strips (BioMérieux, Marcy L’Etoile, France)	Not applicable.	Hyphae detected by optical microscopy. Germ-tube formation assessment. Biofilm formation assessed by crystal violet assay.	Vaginal samples.	The germ-tube formation rate and the biofilm formation ability of strains in the RVVC group were significantly greater than those of strains in the VVC and healthy groups. The antifungal susceptibility of strains in the VVC group was significantly lower than that of strains in the RVVC and healthy groups.	Focused only on *C. albicans* virulence. without analyzing host factors; small sample size; limited geographic scope.	Fair
Rosentul D.C. et al., 2014 [27]	Case–control study	France, Netherlands, and USA	Assess the impact of SNPs in the genes coding for *DECTIN-1, CARD9, TLR1, TLR2*, and *TLR4* on susceptibility to RVVC.	382 women of at least 18 years of age: 119 women with RVVC and 263 healthy women.	Healthy women.	Not specified.	Not applicable.	Gene polymorphisms identified by qPCR. Computational methods used to predict the impact of the Pro631His polymorphism on the TLR2 protein.	Cytokine stimulation functional assays: PBMNs isolated from healthy volunteers and incubated with *C. albicans* blastoconidia. Cytokines measured by ELISA.	Blood samples.	A single *TLR2* polymorphism was associated with a 2.705-fold increase in susceptibility to RVVC. The *TLR2* rs5743704 SNP had deleterious effects on protein function, and it was associated with decreased levels of IL-17 and IFN-γ upon stimulation of PBMNs with *C. albicans.*	Cytokine stimulation assay part of the study: only two individuals in the selected group were heterozygous for the variant allele.	Fair
Tian J. et al., 2021 [18]	Prospective cohort study	China	Determine whether the recurrence of VVC resulted from relapse caused by the same pathogen or reinfection caused by a new pathogen. Compare the genotypes of C. *albicans* from RVVC, VVC, and healthy volunteers to research the correlation between RVVC and *C. albicans* genotypes.	113 women: 44 women with RVVC, 40 VVC patients, and 29 healthy women.	VVC patients and healthy women—genotypes of multiple isolates from the same patient.	*C. albicans* identified by VITEK MS automatic microbial mass spectrometry system.	Not applicable.	*C. albicans* genotyping by MLST.	Not applicable.	Vaginal samples	.The results showed that 59.1% of the patients suffered a relapse, whereas the etiology of the remaining 40.9% of patients was reinfection. Regarding correlation between sources of isolates and genotypes, the differences in results between relapse patients and patients in other groups were statistically significant, but no significant differences were found between the two infectious periods in re-infection patients, VVC patients, and healthy volunteers.	The cause of RVVC was only focused on *Candida* spp. Genotyping; the relationship between the host and RVVC was not assessed; small sample size; limited geographic scope.	Good
Usluogullari B. et al., 2014 [28]	Case–control study	Turkey	Investigate whether Human the *Dectin-1* Y238X Gene Polymorphism plays a role in RVVC pathogenesis.	100 women: 50 women with RVVC and 50 healthy women.	Healthy women.	Diagnosis of candidiasis confirmed using vaginal discharge cultures.	Not applicable.	PCR products sequenced using an ABI-310 sequencer to determine Dectin-1 genotypes.	Not applicable.	Blood samples.	When *Dectin-1* genotypes were compared, there was no significant difference between the two groups (RVVC and controls).	Different polymorphisms in the *Dectin-1* locus were not analyzed; small sample size; limited geographic scope.	Fair

Abbreviations: ASMR-PCR = amplification-refractory mutation system–PCR; CFUs = colony forming units; CLSI = Clinical and Laboratory Standard Institute; DST = diploid sequence type; ELISA = enzyme-linked immunosorbent assay; FISH = fluorescent in situ hybridization; IL = interleukin; ITS = internal transcribed spacer; MBL = mannose-binding lectin; MLST = multilocus sequence typing; NLRP3 = pyrin domain-containing protein 3; PCR = polymerase chain reaction; PCR-RFLP = PCR–restriction fragment length polymorphism; PBMCs = peripheral blood mononuclear cells; PMNs = polymorphonuclear neutrophils; qPCR = quantitative polymerase chain reaction; RVVC = recurrent vulvovaginal candidosis; SDA = Sabouraud dextrose agar; SNPs = single-nucleotide polymorphisms; VNTR = variable-number tandem repeat; VVC = vulvovaginal candidosis.

**Table 2 jof-11-00357-t002:** Antifungal susceptibility tests of *Candida albicans* in patients with RVV.

	Ge G. et al., 2022 [17]				Li X. et al., 2022 [26]				Consuegra-Asprilla J.M. et al., 2024 [20]			
	S	SDD	I	R	S	SDD	I	R	S	SDD	I	R
Antifungal drug	*n*, (%)	*n*, (%)	*n*, (%)	*n*, (%)	*n*, (%)	*n*, (%)	*n*, (%)	*n*, (%)	*n*, (%)	*n*, (%)	*n*, (%)	*n*, (%)
Amphotericin B	2 (11.1)	-	n.a.	16 (88.9)	58 (100)	n.a.	n.a.	-	36 (96.8)	-	-	1 (3.2)
Caspofungin	n.a.	n.a.	n.a.	n.a.	n.a.	n.a.	n.a.	n.a.	37 (100)	-	-	-
Fluconazole	15 (83.3)	2 (11.1)	n.a.	1 (5.6)	52 (89.7)	n.a.	n.a.	6 (10.3)	31 (84.4)	2 (6.2)		4 (9.4)
Itraconazole	11 (61.1)	7 (38.9)	n.a.	-	56 (96.6)	n.a.	n.a.	2 (3.4)	n.a.	n.a.	n.a.	n.a.
Ketoconazole	12 (66.7)	5 (27.8)	n.a.	1 (5.5)	n.a.	n.a.	n.a.	n.a.	n.a.	n.a.	n.a.	n.a.
Micafungin	20 (100)	-	n.a.	-	n.a.	n.a.	n.a.	n.a.	37 (100)	-	-	-
Posaconazole	18 (100)	-	n.a.	-	n.a.	n.a.	n.a.	n.a.	n.a.	n.a.	n.a.	n.a.
Terbinafine	-	-	n.a.	-	n.a.	n.a.	n.a.	n.a.	n.a.	n.a.	n.a.	n.a.
Voriconazole	16 (89.0)	1 (5.5)	n.a.	1 (5.5)	53 (91.4)	n.a.	n.a.	5 (8.6)	32 (87.5)		3 (9.4)	2 (3.1)
5-Flucytosine	n.a.	n.a.	n.a.	n.a.	58 (100)	n.a.	n.a.	-	36 (96.8)	-	-	1 (3.2)

Abbreviations: I = intermediate susceptibility; n.a. = not applicable; R = resistant; SDD = susceptible dose-dependent; S = susceptible.

## Data Availability

Data are contained within the article and Appendix A.

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
