# Peer review of "Recurrent Vulvovaginal Candidosis and Its Underlying Mechanisms: A Systematic Review"

_jof, 2025, doi:10.3390/jof11050357_

Round 1
Reviewer 1 Report
Comments to the authors:
The paper is very interesting in the field. Authors presented an important systematic review describing the underlaying mechanisms in RVVC. The manuscript its well written; minor issues need to be dealt with prior to publication:
Comments:
Please update the reference number 2 where indicate the number of VVC episodes per year (Workowski et al, Sexually transmitted Infections Treatment Guidelines, 2021. MMWR Recomm Rep 70:1-187, BOX 4).
For me, the quality assessment is unclear, given that only one study meets all the criteria for a "good" classification. However, when reviewing several of the other studies, many of them have well-defined criteria that could reclassify this aspect.
It would be important to include a table or figure indicating the main Candida species and their respective susceptibility tests (from the total size of the different studies). Also, indicate whether virulence factors such as gene expression or biofilm formation, among others, were evaluated.
Other relevant information should be included, such as underlying diseases other than those included in the inclusion and exclusion criteria, determining the average age of the total number of patients evaluated, etc.
Comments to the authors:
The paper is very interesting in the field. Authors presented an important systematic review describing the underlaying mechanisms in RVVC. The manuscript its well written; minor issues need to be dealt with prior to publication:
Comments:
Please update the reference number 2 where indicate the number of VVC episodes per year (Workowski et al, Sexually transmitted Infections Treatment Guidelines, 2021. MMWR Recomm Rep 70:1-187, BOX 4).
For me, the quality assessment is unclear, given that only one study meets all the criteria for a "good" classification. However, when reviewing several of the other studies, many of them have well-defined criteria that could reclassify this aspect.
It would be important to include a table or figure indicating the main Candida species and their respective susceptibility tests (from the total size of the different studies). Also, indicate whether virulence factors such as gene expression or biofilm formation, among others, were evaluated.
Other relevant information should be included, such as underlying diseases other than those included in the inclusion and exclusion criteria, determining the average age of the total number of patients evaluated, etc.
Reviewer 2 Report
Dear editor and authors
I have received the manuscript “Jof-3573880” entitled “Recurrent Vulvovaginal Candidosis and its Underlying Mechanisms: A Systematic Review” to review in Mar/Apr 2025.
This is an interesting review about RVVC highlighting immunological aspects of patients, antifungal treatment and microbiological characteristics of yeasts associated with this condition. The authors were able to achieve their objectives after an extensive review of articles published in the last 10 years.
I made some recommendations, mainly related to scientific nomenclature. The genus Candida must be followed by a specific species (e.g. Candida albicans or C. albicans). Or by Candida species (or even Candida spp.). The words “species or spp.” cannot be italicized.
It is not correct to say “non-albicans”. The appropriate way is non-C. albicans. The authors presented non-albicans species; non-albicans Candida species. This must be presented in a standardized manner.
I am reporting this manuscript as “minor revisions”. In my opinion, this manuscript is very close to being accepted for publication in the JoF.
I attached some more comments.
attached

Reviewer 3 Report
I appreciate the work and time you spend on writing this manuscript. I appreciate the poverview you give of the literature, however, this alone does not make the work new and insightful. You can lift the work up, by give a more in depth discussion of your findings and how it aligns or not with the in vivo research and patient studies. The Th17 response is an interesting example of how you can take your discussion to a higher level. Same regarding the lack of clearnace by PMN's, and place it in a broader light. The study would also benefit from your opinion on how the findings discussed in your paper could affect the treatment of RVVC.
See above.
Author Response
Comment 1:I appreciate the work and time you spend on writing this manuscript. I appreciate the powerview you give of the literature, however, this alone does not make the work new and insightful. You can lift the work up, by give a more in depth discussion of your findings and how it aligns or not with the in vivo research and patient studies. The Th17 response is an interesting example of how you can take your discussion to a higher level. Same regarding the lack of clearnace by PMN's, and place it in a broader light.
Response 1: We appreciate the suggestions that improved the article. We have added the following to the discussion:
“It is worth noting that IL-17/Th 17 plays a crucial role in neutrophil recruitment and activation. Candida killing by mononuclear leukocytes from Th17 inhibitor recipients with a history of candidiasis was impaired compared to healthy controls, in a real-world observational study.” - lines 489-492 (page 19); Reference 37 (Davidson, L., et al., Risk of candidiasis associated with interleukin-17 inhibitors: A real-world observational study of multiple independent sources. Lancet Reg Health Eur, 2022. 13: p. 100266.)
“L-17 inhibitors are currently used to reduce inflammation and tissue injury associated with chronic inflammatory diseases (like psoriasis and ankylosing spondylitis) by targeting T helper -17 pathway. These drugs are associated with an increased risk of candidiasis, including VVC. In fact, observational data and clinical trials show the increased incidence of candidiasis in individuals treated with IL-17 inhibitors.” - lines 514 - 519 (page 19); References 37 and 41 (Davidson, L., et al., Risk of candidiasis associated with interleukin-17 inhibitors: A real-world observational study of multiple independent sources. Lancet Reg Health Eur, 2022. 13: p. 100266; Bilal, H., et al., Risk of candidiasis associated with interleukin-17 inhibitors: Implications and management. Mycology, 2024.
649 15(1): p. 30-44)
“Furthermore, the immune response against Candida species is compromised when IL-17 is also inhibited through genetic knockout” - lines 520, 521 (page 20); Reference 42 (Puel, A., et al., Chronic mucocutaneous candidiasis in humans with inborn errors of interleukin-17 immunity. Science, 2011. 332(6025): p. 65-8).
Comment 2: The study would also benefit from your opinion on how the findings discussed in your paper could affect the treatment of RVVC.
Response 2: We have added the following to the discussion:
“In this era of antifungal resistance, the search for individualized immunomodulatory treatment that enhances and improves the host's immunity in the fight against Candida infections is an urgent need. Anti-Candida vaccines are under development, and promising results are awaited. With advances in the understanding of the immunopathogenesis of RVVC and host-related factors, this challenge is becoming more tangible and promising.” - lines 522-527 (page 20).
Reviewer 4 Report
The manuscript is well-written and addresses the highly pertinent topic of Recurrent Vulvovaginal Candidiasis (RVVC), which represents a significant challenge for clinicians, scientists, and patients. The review presented is comprehensive and effectively meets the proposed objectives. I have only a few minor suggestions for the manuscript's further improvement:
I have only a few minor suggestions for the manuscript's further improvement:
- Candida glabrata has recently been reclassified into the genus Nakaseomyces. This taxonomic change should be mentioned upon the first appearance of the species name, along with an appropriate reference (page 1). This clarification is necessary even if the originally cited articles used the former nomenclature.
- Please insert a reference on page 2, line 3.
- The abbreviation "spp." should not be italicized. Please remove italics from this abbreviation throughout the manuscript.
- Table 1 is too long, it contains excessive information. Please consider revising its layout, perhaps by summarizing the data more effectively or by transferring some details from the table into the main text.
- Acronyms used in the table, such as "DST" (Diploid Sequency Type), should be defined in the table footnote.
Round 2
Reviewer 3 Report
It is another review on a complicated matter. Thank you for making the changes.
Thanks for making the requested changes.